# Taurine Attenuates Catabolic Processes Related to the Onset of Sarcopenia

**DOI:** 10.3390/ijms21228865

**Published:** 2020-11-23

**Authors:** Alessandra Barbiera, Silvia Sorrentino, Elisa Lepore, Andrea Carfì, Gigliola Sica, Gabriella Dobrowolny, Bianca Maria Scicchitano

**Affiliations:** 1Sezione di Istologia ed Embriologia, Dipartimento di Scienze della Vita e Sanità Pubblica, Fondazione Policlinico Universitario A. Gemelli IRCCS, 00168 Roma, Italy; Alessandra.Barbiera@unicatt.it (A.B.); Silvia.Sorrentino@unicatt.it (S.S.); andreacarfi39@ymail.com (A.C.); Gigliola.Sica@unicatt.it (G.S.); 2DAHFMO-Unità di Istologia ed Embriologia Medica, Sapienza Università di Roma, 00161 Roma, Italy; elisa.lepore@uniroma1.it (E.L.); Gabriella.Dobrowolny@uniroma1.it (G.D.)

**Keywords:** skeletal muscle, aging, nutrition, inflammation, antioxidants, autophagy

## Abstract

Sarcopenia that occurs with advancing age is characterized by a gradual loss of muscle protein component due to the activation of catabolic pathways, increased level of inflammation, and mitochondrial dysfunction. Experimental evidence demonstrates that several physio-pathological processes involved in the onset of sarcopenia may be counteracted by the intake of specific amino acids or antioxidant molecules, suggesting that diet may represent an effective strategy for improving the anabolic response of muscle during aging. The non-essential amino acid taurine is highly expressed in several mammalian tissues, including skeletal muscle where it is involved in the ion channel regulation, in the modulation of intracellular calcium concentration, and where it plays an important role as an antioxidant and anti-inflammatory factor. Here, with the purpose to reproduce the chronic low-grade inflammation characteristics of senescent muscle in an in vitro system, we exploited the role of Tumor Necrosis Factor α (TNF) and we analyzed the effect of taurine in the modulation of different signaling pathways known to be dysregulated in sarcopenia. We demonstrated that the administration of high levels of taurine in myogenic L6 cells stimulates the differentiation process by downregulating the expression of molecules involved in inflammatory pathways and modulating processes such as autophagy and apoptosis. Although further studies are currently ongoing in our laboratory to better elucidate the molecular mechanisms responsible for the positive effect of taurine on myogenic differentiation, this study suggests that taurine supplementation may represent a strategy to delay the loss of mass and functionality characteristic of senescent muscles.

## 1. Introduction

The progressive loss of skeletal muscle mass and function, often referred to as sarcopenia, is the result of multiple factors, including malnutrition, physical inactivity, and altered hormone levels that may all contribute to deregulation of the molecular milieu of the aged muscle, ultimately leading to muscle wasting and frailty [1,2]. During aging, a chronic low-grade inflammatory state known as “inflammaging” occurs, characterized by an increased systemic concentration of proinflammatory cytokines [3]. In particular, Tumor Necrosis Factor α (TNF) and Interleukin-6 (IL-6) have been implicated in the inflammatory response in skeletal muscle, and an age-related increase of these inflammatory markers is thought to contribute to the sarcopenic phenotype [4,5,6]. Although the molecular mechanisms involved in the interaction between inflammaging and muscle loss have not yet been understood, recent findings demonstrate that the increased levels of low-grade inflammation may favor dysregulation between synthesis and protein degradation [2,7,8].

Dietary protein or amino acid ingestion is one of the most effective physiological stimuli to enhance muscle net protein balance, due to the amino acid-dependent stimulation of protein synthesis [9,10]. However, the diminished food intake typical of older adults often results in an attenuated muscle protein synthetic response when compared to that of younger persons. This “anabolic resistance” of muscle protein synthesis seems to be the major underlying mechanism of the loss of muscle mass in old individuals and suggests that nutrition may represent an effective strategy aimed to improve the anabolic response of the muscle in the elderly [11].

One of the most important signaling known to control protein turnover in skeletal muscle is mediated by the phosphatidylinositol 3-kinase (PI3K). The activation of the PI3K pathway triggers a cascade of intracellular effectors, including the serine/threonine kinase Akt, which controls both protein synthesis, by activating the mammalian target of rapamycin (mTOR) kinase, and protein degradation by repressing the members of the fork-head family of transcription factors (FoxO) [12,13,14]. Nuclear translocation and activation of FoxO transcription factors induce an increased expression of key regulating factors of the muscle-specific ubiquitin-ligases, atrogin-1 (MAFbx), and Muscle RING Finger 1 (MuRF1) that are over-expressed in the elderly and are involved in the onset of sarcopenia [15].

As mentioned above, aging is generally associated with a chronic state of slightly increased plasma levels of pro-inflammatory mediators, which could potentially induce muscle catabolism by deregulating the metabolism of amino acids [16]. Pro-inflammatory cytokines such as TNF and IL-6 may accelerate frailty either directly, by promoting muscle protein degradation, or indirectly, by affecting important metabolic signaling pathways [17]. In particular, TNF is able to activate the nuclear factor kappa B (NF-kB) pathway in several cell types, including muscle cells where its effect is associated with the onset of sarcopenia [18,19]. Inactive NF-kB is bound to a family of inhibitory proteins called IkB, but under the TNF-dependent stimulation of an IkB kinase (IKK) complex, IkB is phosphorylated, triggering its ubiquitination and proteasome degradation. This leads to nuclear translocation of NF-kB and activation of NF-kB-mediated gene transcription [20].

In skeletal muscle, NF-kB activation is involved in different processes, including the stimulation of protein catabolism, and the upregulation of inflammatory cytokines, thus establishing a positive feedback loop resulting in the over-stimulation of NF-κB and the development of muscular abnormalities [19]. Besides, it has been demonstrated that NF-kB can induce the expression of FoxO transcription factors by modulating the Akt/mTOR signaling cascade [21,22], ultimately leading to skeletal muscle loss and functional impairment.

Autophagy, a major protein turnover pathway by which cellular components are delivered into the lysosomes for degradation and recycling, is also constitutively active in skeletal muscle [23]. Experimental evidence reveals that during aging, autophagic processes may be dysregulated in muscle and this reduced autophagic capacity may be related to the loss of skeletal muscle mass and functionality in both humans and rodents [24]. The decreased autophagy gene expression [24,25], the lower protein levels of autophagy core components and regulators [26,27], and the modulation of mTOR signaling [14] are all suggested as possible contributors to the progressive age-related decline of the autophagic system.

Taurine, a derivative of cysteine, is a sulfur-containing semi-essential amino acid not incorporated into proteins. In mammalian, taurine is the amino acid prevalent in excitable tissues where it plays an important role in several biological processes including ion channel regulation and modulation of intracellular calcium concentration, and where it plays an important role as an antioxidant and anti-inflammatory factor [6,28]. Dietary compounds, in particular fish and meat, are the main source of taurine [29], whose concentration inside the cells is guaranteed by the presence of an active transporter (TauT), ubiquitously expressed in many mammalian tissues, that can concentrate taurine within cells against its gradient. The evidence of a physiological role for this amino acid is suggested by the fact that several animal species, such as felines and foxes, that are unable to synthesize taurine, are particularly susceptible to a deficient state developing several pathophysiological conditions, such as retinal degeneration, cardiomyopathy, and reproductive defects [28,29,30]. It is known that in several tissues, taurine level decreases during aging. Besides, taurine-depleted skeletal muscle exhibits several abnormalities in its morphology and function, resembling those that occur during aging [31].

Here, we used an in vitro experimental model to demonstrate that the treatment of myogenic L6 cells with high levels of taurine stimulates the differentiation process, downregulates the expression of molecules involved in inflammatory pathways, and modulates the activation of processes such as autophagy and apoptosis. Although further in vivo studies are currently ongoing in our laboratory to better elucidate the molecular mechanisms responsible for the positive effect of taurine on myogenic homeostasis, this study suggests that taurine supplementation may represent a strategy to delay the loss of mass and functionality characteristics of senescent muscles.

## 2. Results

### 2.1. Taurine Counteracts the Negative Effects of TNF on L6 Cell Differentiation

To investigate the effect of taurine on myogenic differentiation, L6 cells cultured in a serum-free medium were exposed to 200 μM taurine, 15 ng/mL TNF, or their combination. TNF was used to reproduce the chronic low-grade inflammation characteristic of senescent muscle in an in vitro system [5]. After 7 days, the differentiation was assessed by morphological analysis of the cell fusion process (Figure 1A). While the different treatments did not significantly affect the total number of nuclei, the taurine treatment stimulates the differentiation of L6 cells as confirmed by the presence of large myotubes containing an average of 35 nuclei/fiber, compared to the control cultures (Figure 1B,C). Interestingly, taurine-treated L6 cells show a characteristic organization of the nuclei, forming nuclear rings, which represent a morphological marker of muscle hypertrophy/maturation in vitro [32,33]. As expected, the differentiation of L6 cells is hampered by the presence of TNF alone, as confirmed by the very low percentage of fusion that reaches only 5% in this condition, while the cultures that received the combined treatment with taurine and TNF show a level of differentiation comparable to that of control (Figure 1A,C). These results demonstrate that taurine can counteract the negative effect of TNF on myogenic differentiation.

The effect of taurine on myogenesis was also evaluated by analyzing the expression of one of the key regulators of the differentiation program such as Myosin Heavy Chain (MHC). Immunofluorescence analysis shows that MHC expression is low in the few and thin myotubes observed in the control, while its expression dramatically increases in the multinucleated myotubes present in taurine-treated cultures. As expected, TNF dramatically inhibits myoblast fusion and downregulates MHC expression; however, when the cells were treated with taurine and TNF together, MHC expression is highly detectable in multinucleated myotubes, demonstrating a significant effect of taurine in counteracting the TNF negative effect on myogenic differentiation (Figure 1D). In addition, Western Blot analysis of MHC expression performed after 3 days of the treatments indicated above reveals that at this time although the taurine effect is not yet significant compared to the control cultures, taurine is able to counteract the negative effect of TNF on MHC expression levels (Figure 1E,F).

### 2.2. PI3K/Akt Signaling Is Involved in Taurine-Dependent Myogenic Differentiation of L6 Cells

One of the major pathways that have been shown to have a prominent role in increased protein synthesis associated with muscle differentiation/hypertrophy is the PI3K/Akt signaling [34,35]. To verify the involvement of PI3K/Akt signaling in the taurine effect on myogenic differentiation, we used a genetic approach, i.d. Akt1 silencing by transiently transfected L6 cells with siRNA against Akt1. Western blot analysis demonstrates that the strong upregulation of Akt expression observed in taurine treated samples is dramatically downregulated by the Akt ablation (Figure 2A,B). Moreover, since mTOR and MEF2 proteins are implicated as downstream targets of PI3K/Akt signaling as key factors in the control of muscle gene expression, we analyzed by Western Blotting their protein expression in the same conditions. Figure 2A,C,D show that both mTOR and MEF2 are upregulated in the vehicle-transfected taurine treated cells while the silencing of Akt1 is associated with a profound reduction in their expression levels. These results suggest that the Akt signaling network is crucial for myoblast differentiation induced by taurine.

### 2.3. Taurine Counteracts the TNF-Dependent Activation of NF-κB Expression

High levels of inflammatory cytokines such as TNF were reported to negatively affect skeletal muscle protein metabolism by the suppression of the PI3K/Akt pathway through the activation of NF-κB transcription factors, ultimately leading to muscle wasting [36,37,38].

Here we investigated whether the positive effect of taurine on myogenic differentiation was mediated by the modulation of inflammatory-related pathways. To this purpose, we treated L6 cells with taurine in the presence or in absence of TNF for three days and analyzed phospho-NF-κB expression. We observed, by immunofluorescence analysis, that phospho-NF-κB is strongly upregulated in the nuclei of L6 cells after TNF treatment while its expression appears low and diffuse in control and when the cells received taurine alone or in combination with TNF (Figure 3A). Western Blot analyses confirmed the role of TNF in promoting phospho-NF-κB upregulation and showed the concomitant downregulation of the NF-kB inhibitory factor IκBα (Figure 3B–E). Differently, phospho-NF-κB expression was significantly reduced in muscle treated with both taurine and taurine + TNF, demonstrating that taurine attenuates the effects of TNF on inflammation (Figure 3B,C). Besides, we evaluated by Real Time Polymerase Chain Reaction (PCR) analysis the expression levels of one of the most relevant inflammatory molecules such as TGF-β, confirming a role for taurine in the down-regulation of inflammatory processes (Figure 3F).

### 2.4. Modulation of the Autophagic Pathway by Taurine Treatment

To investigate the role of taurine in the regulation of the autophagy process, we initially treated L6 cells with taurine in the presence or absence of TNF for three days and analyzed the expression levels of proteins known to play a crucial role in the formation of autophagosomes. In particular, we evaluated the expression of LC3, involved in autophagosome formation, and Atg9, important for the fusion of the autophagosome with the lysosome. Pre-LC3 is processed to its cytosolic form, LC3-I, which is then activated and lipidated to its membrane-bound form, LC3-II, localized to the preautophagosome structure and autophagosomes. As shown in Figure 4A,B, we detected high levels of LC3II/I ratio (that has been suggested as a more precise biochemical marker revealing ongoing autophagy [39,40]) in the control and when the cells were treated with TNF alone, but LC3II/I ratio decreased in the presence of taurine and of taurine + TNF.

Besides, the integral membrane protein Atg9, strongly upregulated by TNF treatment, is dramatically reduced when the cells received taurine either in the presence and in the absence of TNF (Figure 4C,D). The results were supported by the immunofluorescence analysis shown in Figure 4E,F demonstrating that taurine can decrease the expression of the autophagic markers LC3 and Atg9 even in the presence of TNF.

### 2.5. Taurine Affects the Apoptotic Pathway through the Modulation of Caspase 3 Expression

It has been demonstrated that autophagy is implicated in the cell death process in response to cytotoxic stimuli [41] and, since in certain circumstances both apoptosis and autophagy can occur concomitantly in the same cells, we verified whether taurine can exert its effect in the modulation of TNF-dependent induction of apoptotic pathway. Indeed, the increase in circulating concentrations of the cytokines, such as TNF, may initiate pro-apoptotic signaling, stimulating cleavage and activation of the executioner caspases directly linked to pro-apoptotic changes.

To this purpose, we treated L6 cells with taurine in the presence or the absence of TNF and we evaluated by Western Blot analysis the expression levels of caspase 3. As shown in Figure 5A,B, the increased levels of the cleaved isoform of caspase 3 detected in TNF treated cells is significantly decreased by taurine both in the presence and in the absence of TNF.

## 3. Discussion

In the present paper, we investigated the effect of taurine on myogenic differentiation and we exploited the role played by the inflammatory cytokine TNF to reproduce the level of chronic inflammation characteristic of senescent muscle in an in vitro system. The choice to use taurine is based on the fact that a marked decrease in taurine content has been observed in human muscle specimens with age [42] and, besides sarcopenia, skeletal muscle of aged rats develops features that are overlapping those observed in taurine depleted muscles, i.e., a marked decrease in gCl and a change in calcium homeostasis [43,44].

Our results reveal that taurine induces the differentiation of L6 cells as demonstrated by the marked increase in myoblast fusion index and by the expression of one of the main markers of the differentiation process such as MHC. Interestingly, the positive myogenic effect of taurine is maintained even when the cells simultaneously received taurine and TNF. Besides, a characteristic central disposition of the nuclei was the evidence of the hypertrophic effect of taurine in these cells [45,46]. These results are in agreement with data reported by Miyazaki T. et al. demonstrating that taurine treatment significantly enhanced myogenic differentiation in a dose-dependent manner, while this effect was abrogated by inhibiting taurine transport and Ca^2+^ signaling [47].

We demonstrated that the stimulation of the Akt/mTOR pathway plays a crucial role in the mechanism involved in the taurine effect on myogenic differentiation/growth. Indeed, the abrogation of Akt expression strongly decreased the taurine dependent upregulation of mTOR and MEF2 transcription factor, which is involved in the activation of muscle-specific genes and is upregulated during myogenesis [48]. Notably, an upregulation of taurine transporter (TauT) mRNA and protein expression has been demonstrated after the initiation of myogenesis in C2C12 myoblasts, and a MEF2 binding site is present in the promoter region of TauT [49]. These data, together with the fact that muscle taurine concentration is several times higher in the fetal and neonatal states than in mature mammals [50], strongly suggest that taurine may participate in myogenic differentiation during muscle development and regeneration.

It is well established that TNF increases muscle wasting [51] and it is also known to be a potent activator of NF-κB transcription factor, which is involved in the modulation of different processes in several tissues, including muscle, controlling, among others, inflammation, apoptosis, and autophagy signaling pathways [18,20,52]. In this paper, we investigated whether taurine counteracts the negative effect of TNF on myogenic differentiation by the modulation of NF-κB signaling. Generally, in unstimulated cells, NF-κB is retained in the cytoplasm by binding to IκB proteins. In response to specific stimuli, such as inflammatory cytokines, proteasome-dependent degradation of IκB allows the translocation of NF-κB to the nucleus, where it activates different cellular responses [53]. We demonstrated that the strong nuclear expression of NF-κB observed in the presence of TNF dramatically decreases by taurine treatment and, on the contrary, the expression of IκB, high in the extracts of the cells treated with taurine, is significantly down-regulated by TNF treatment. Besides, taurine also downregulates the expression of another relevant inflammatory marker such as TGF-β. TGF-β is known to synergize with TNF or other cytokines, suggesting a convergence of the two pathways at common target genes. In particular, NF-κB can be activated by TGF-β and mediate transcription of target genes in a variety of cell types [54]. This result is consistent with the role of taurine as an anti-inflammatory molecule [53]. Indeed, recent findings demonstrated that taurine can scavenge HOCl produced during inflammation to form the more stable and less toxic Tau-Cl, which exerts its effect by inhibiting the production of inflammatory mediators through a mechanism that, at least in part, involves the inhibition of NF-κB activation [6].

Mitochondrial dysfunction may represent a crucial event in the onset of sarcopenia. Indeed, alteration in the mitochondrial network during aging has potentially deleterious consequences for the maintenance of muscle mass and function and may contribute to increased mitochondrial-derived ROS production, which may result in altered apoptotic and autophagic processes [55,56]. TNF has been shown to stimulate ROS production by mitochondria and, in skeletal muscle, endogenous ROS mediate NF-κB activation that may result in inhibition of myogenesis [57,58,59]. Here, we analyzed the expression levels of key proteins that are involved in the formation of autophagosomes (Atg9, LC3) and the fusion of the autophagosome with the lysosome, demonstrating that in L6 cells taurine treatment decreases the high levels of LC3II/I and Atg9 observed in control and in the presence of TNF. It has to be considered that in our experiments L6 cells were cultured in a serum-free medium that allows us to study the specific role of taurine, avoiding potential synergistic or additive effects of factors generally present in the serum. This is particularly relevant for myogenic cells whose differentiation is classically induced by decreasing the concentration of serum in the culture medium [60,61]. Since nutrients are the main regulators of autophagy [62], it is not surprising that this culture condition may induce the autophagy process even in control culture masking the effect of TNF. However, taurine can decrease the expression of autophagy markers in these conditions. This effect is consistent with the role of taurine in the attenuation of oxidative stress and the improvement of mitochondrial functions [63,64,65,66,67]. Although the molecular signaling involved in the antioxidant effect of taurine is not still fully elucidated, recent findings suggest that taurine increases antioxidant levels through increased mitochondrial electron transport chain activity [63,64,67].

As previously mentioned, TNF-dependent induction of oxidative damage to mitochondrial components increases ROS generation and may lead to apoptotic events [68]. Upregulation of myocytes apoptosis has been shown in animal models of premature aging as well as in aged rodents and humans [69,70].

Caspases represent key players in the molecular mechanisms involved in the stimulation of apoptotic processes [71]. In particular, the activation of the effector caspase 3 seems to be involved in an initial step, leading to an acceleration of muscle proteolysis in catabolic conditions [72,73]. Our results show that, in L6 cells, taurine treatment counteracts the effect of TNF on the cleaved caspase 3 expression levels, demonstrating an anti-apoptotic role of taurine that may protect the degradation of contractile proteins of the muscle. Indeed, taurine treatment is able to counteract the upregulation of atrophic markers such as Murf1, strongly upregulated after TNF treatment (data not shown).

## 4. Materials and Methods

### 4.1. Cell Culture and Treatments

L6 rat myogenic cells were seeded at the density of 25,000/cm^2^ and cultured in growth medium (GM) consisting of: Dulbecco’s modified Eagle’s medium (DMEM) supplemented with 10% heat-inactivated fetal bovine serum (FBS), 2 mM L-glutamine, 100 U/mL penicillin, and 100 μg/mL streptomycin. The cells were incubated at 37 °C in a humidified atmosphere with 5% carbon dioxide and 95% air. Twenty-four hours after plating, cultures were washed with Phosphate Buffer Saline (PBS) and shifted to low-serum medium consisting of DMEM supplemented with 1% fatty acid-free bovine serum albumin (BSA, #A9418 Sigma, St. Louis, MO, USA) [42] and treated with 200 μM Taurine (#T8691 Sigma, St. Louis, MO, USA) and/or 15 ng/mL recombinant mouse TNF-α (#11271156001; Roche, Indianapolis, IN, USA) at different times as described below.

### 4.2. Measurement of Myoblast Fusion and Growth

L6 myoblasts were plated at a density of 25,000/cm^2^ and cultured for 24 h in GM. Cells were then shifted in serum-free medium and treated as described above. Cell fusion was evaluated by May-Grunwald-Giemsa staining (MGG quick stain, Bio-Optica, Milan, Italy) after 7 days of treatment. Cells were considered fused only if cytoplasmic continuity and at least three nuclei were present in each myotube. The ratio between the numbers of nuclei in myotubes versus the total number of nuclei per microscopic field was expressed as the percentage of fusion. Each experimental point represented in the graphs is the mean ± SD of the counts from three independent experiments.

### 4.3. Immunofluorescence Analyses

L6 myoblasts were plated on glass circle coverslips at a density of 25,000/cm^2^ and cultured for 24 h in GM. Then, the cells were shifted in a serum-free medium and treated for 3 or 7 days as indicated. Subsequently, the cells were fixed in ice-cold acetone: methanol (1:1) for 20 min at −20 °C, permeabilized with 0.01% Triton X-100 in PBS for 3 min and blocked for 30 min with 0.3% Triton X-100 and 5% donkey serum in PBS. Cells were then incubated overnight at 4 °C with the selected primary antibodies at the appropriate dilution in PBS containing 1.5% donkey serum. Antibodies used were as follows: rabbit monoclonal anti-LC3A/B (1:100, #12741, Cell Signaling Technology, Danvers, MA, USA), rabbit monoclonal anti-Atg9A (1:100, #13509, Cell Signaling Technology, Danvers, MA, USA), rabbit monoclonal anti-phospho- NF-κB p65 (Ser468) (1:100, #3039, Cell Signaling Technology), and mouse monoclonal anti-MHC (MF20) (1:100, #AB_2147781, Developmental Hybridoma-bank, University of Iowa, IA,). Cells were then washed three times in PBS and incubated at room temperature for 60 min with appropriate secondary antibodies: AlexaFluor594-conjugated anti-mouse 1:1000 (Molecular Probes, Eugene, OR, USA, #A21203) and AlexaFluor488-conjugated anti-rabbit 1:1000 (Molecular Probes, Eugene, OR, USA, #A21206) in PBS containing 1.5% donkey serum. The coverslips were mounted with ProLong™ Gold Antifade Mountant with DAPI (Thermo Fisher Scientific, Waltham, MA, USA, #P36935), examined with an Olympus BX53 fluorescent microscope (Olympus, Tokyo, Japan) and captured with ISCapture software (Tucsen Photonics, Fujian, China).

### 4.4. siRNA Transfection

L6 myoblasts were plated at a density of 25,000/cm^2^ and cultured in GM. Cells Transfection with Control and Akt1 siRNAs (TriFECTA Kit DsiRNA Duplex, IDT Technologies) using Mirus TransIT-X2^®^ Transfection Reagent (Mirus Bio Corporation, Madison, WI, USA) were performed according to the manufacturer’s instructions within the recommended reagent/siRNA ratio range. Briefly, 25 nM of each siRNA was diluted in 0.25 mL of Opti-MEM I Reduced Serum Media (Thermo Fisher Scientific, Waltham, MA, USA) and 7.5 μL of Mirus TransIT-X2^®^ Transfection Reagent. The mixture was incubated 30 min at room temperature and then added to each well. After 48 h, cells were washed with PBS, shifted to serum-free medium, and treated for 3 days as described above.

### 4.5. Protein Extraction and Western Blot Analysis

L6 myoblasts were plated on glass circle coverslips at a density of 25,000/cm^2^ and cultured for 24 h in GM. Then, the cells were shifted in a serum-free medium and treated for 3 days as described above. For total homogenates cells were washed in  PBS, harvested, and lysed in Cell Lysis Buffer (Cell Signaling #9803) containing 1 mM PMSF (Cell Signaling #8553) and a complete protease inhibitor cocktail (Cell Signaling #5872) for 30 min at 4 °C. Then the cells were briefly sonicated and the extracts were centrifuged for 10 min at 13,000× *g* in a cold microfuge.

An equal amount of proteins was determined through the Bradford Protein Assay (Bio-Rad Laboratories Inc., Hercules, CA, USA) and Varioskan™ LUX controlled by Thermo Scientific™ SkanIt™ Software for Microplate Readers (Thermo Fisher Scientific, Waltham, MA, USA), according to the manufacturer’s instructions. Proteins were separated by SDS/PAGE (Mini-PROTEAN^®^ TGX™ Precast Protein Gels, or Mini-PROTEAN TGX stain-free precast PAGE gels, Bio-Rad Laboratories Inc., Hercules, CA, USA) and transferred to a nitrocellulose membrane (Trans-Blot^®^ Turbo™ Mini Nitrocellulose Transfer Packs #1704158, Bio-Rad Laboratories Inc., Hercules, CA, USA). Nonspecific binding was blocked in Tris-buffered saline (TBS) (Bio-Rad Laboratories Inc., Hercules, CA, USA) supplemented with 0.1% Tween-20 and containing 5% nonfat dry milk (Bio-Rad Laboratories Inc., Hercules, CA, USA #1706404) for 1h at room temperature. The primary antibodies used are mouse monoclonal anti-MHC (MF20) (1:100, #AB_2147781, Developmental Hybridoma-bank, University of Iowa, IA, USA), mouse monoclonal anti-Caspase-3 (1:1000, #14220, Cell Signaling), rabbit monoclonal anti-LC3A/B (1:1000, #12741, Cell Signaling Technology, Danvers, MA, USA), rabbit monoclonal anti-Atg9A (1:1000, #13509, Cell Signaling Technology, Danvers, MA, USA), rabbit monoclonal anti-Phospho-NF-κB p65 (Ser468) (1:1000, #3039, Cell Signaling Technology, Danvers, MA, USA), rabbit monoclonal anti-IκBa (1:1000, #4812, Cell Signaling Technology, Danvers, MA, USA), rabbit monoclonal anti-mTOR (1:1000, Cell Signaling Technology, Danvers, MA, USA, #2972), rabbit monoclonal anti-Akt (Thr308) (1:1000, Cell Signaling Technology, Danvers, MA, USA, #13038), rabbit polyclonal anti-MEF2 (1:1000, H300, sc-10794, Santa Cruz).

Subsequently, blots were overlaid with the following corresponding secondary antibodies from Bio-Rad Laboratories: Goat anti-Rabbit IgG (1:3000, HRP Conjugate, #1706515) and Goat anti-Mouse IgG (1:3000, HRP Conjugate, #1706516) for 1h at RT. Signals were captured by ChemiDoc™ Imaging System (Bio-Rad Laboratories, Hercules, CA, USA) using an enhanced chemiluminescence system (SuperSignal Chemoluminescent substrate, Thermo Fisher Scientific Inc. Waltham, MA, USA). Densitometric analyses were performed with Image Lab™ Touch Software (Bio-Rad Laboratories). All experiments were carried out in triplicate and representative results are shown in the figures.

### 4.6. Real Time PCR Analysis

Total RNA (1 μg) was reverse-transcribed using the QuantiTect Reverse Transcription Kit (Qiagen). Quantitative PCR was performed using the ABI PRISM 7500 SDS (Applied Biosystems, USA), Taqman universal MMIX II and TaqMan probe (Life Technologies). Quantitative RT-PCR sample value was normalized for the expression of β-Actin mRNA. The relative level for each gene was calculated using the 2--∆∆Ct method (Livak and Schmittgen, 2001) and reported as fold change.

### 4.7. Statistical Analysis

Data are presented as the mean  ±  SD. The Student’s 𝑡-test was used throughout this paper for statistical analyses, assuming equal variance, and *p*-values were calculated based on the 2-tailed test. A *p*-value of <0.05 was considered statistically significant.

## 5. Conclusions

The results reported in this work demonstrate that taurine represents an important myogenic factor able to enhance the differentiation process even in conditions of altered homeostasis. In L6 cells, taurine counteracts the negative effects of TNF by regulating inflammation and modulating autophagic and apoptotic processes. Although further studies are needed to better clarify the molecular mechanisms by which taurine exerts its positive effect on muscle homeostasis, this work suggests that taurine may represent an interesting candidate to counteract the loss of mass and functionality which characterizes sarcopenic muscle.

## Figures and Tables

**Figure 1 ijms-21-08865-f001:**
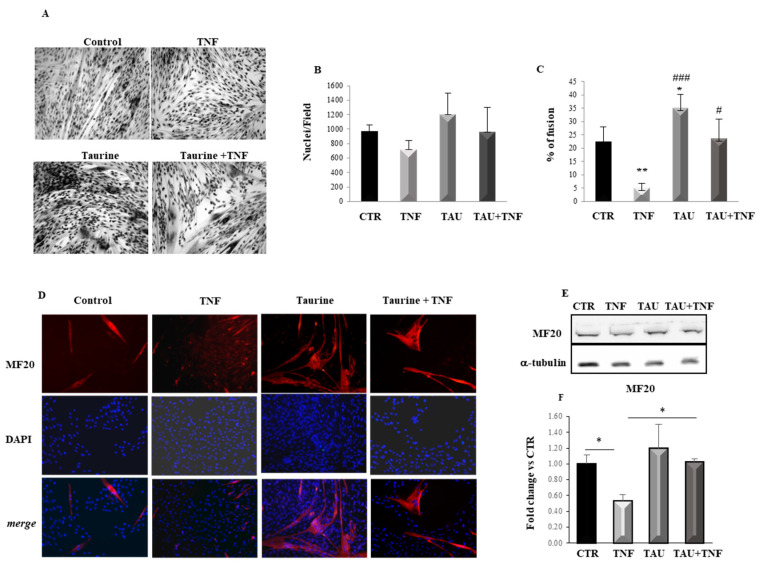
Taurine counteracts the Tumor Necrosis Factor α (TNF)-dependent effect on myogenic differentiation. (**A**) Morphological analysis of L6 cell differentiation evaluated by May-Grunwald Giemsa staining after 7 days of the indicated treatments. (**B**) Diagram showing the total number of nuclei (fused and unfused) per microscopic field in control, TNF-, taurine-, and TNF + taurine-treated cells. (**C**) Diagram showing the ratio between the numbers of nuclei present in myotubes versus the total number of nuclei per microscopic field, expressed as the percentage of fusion. Each experimental point represented in the graphs is the mean ± SD of the counts from three independent experiments. (**D**) Immunofluorescence analysis of L6 cells shows Myosin Heavy Chain (MHC) (MF20) expression in the cultures treated as described above (original magnification × 20). (**E**) Western Blotting of the MHC (MF20) expression performed after 3 days of the indicated treatments. A representative blot is shown. (**F**) Densitometric analyses performed by using anti-alpha-tubulin antibody to verify equal loading of the samples. Statistical analysis was performed using Student’s *t*-test *# *p* < 0.05; ** *p* < 0.01; ### *p* < 0.001. For Figure 1C * vs. Control, # vs. TNF.

**Figure 2 ijms-21-08865-f002:**
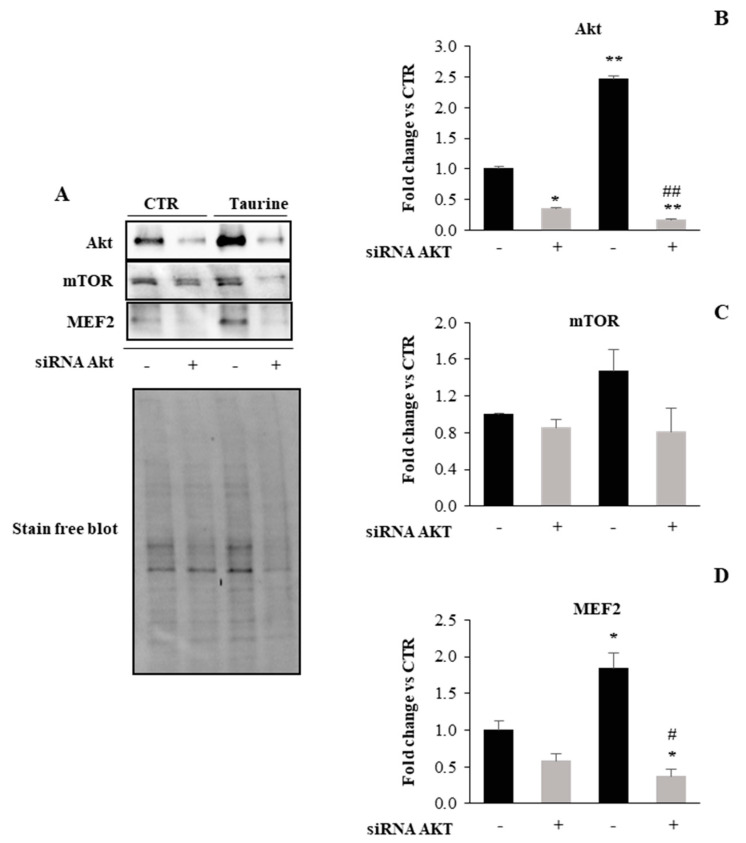
Akt knockdown hampers the effect of taurine on mTOR and MEF2 expression levels. (**A**) Representative Western blots showing Akt, mTOR, and MEF2 protein expression levels in L6 cells extracts treated with taurine and with taurine after silencing of Akt1 gene. (**B**–**D**) Densitometric analyses were achieved using a Stain-free blot to verify the equal loading of the samples. Graphs show mean ± SD presented as arbitrary units relative to control-vehicle-transfected cells. Statistical analysis was performed using Student’s *t*-test. *# *p* < 0.05; **, ## *p* < 0.01. * Vs. control-vehicle transfected cells; # vs. taurine treated cells.

**Figure 3 ijms-21-08865-f003:**
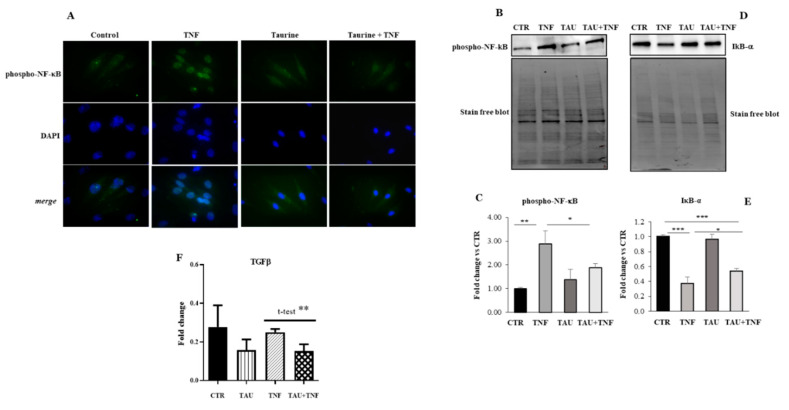
Taurine counteracts the TNF-dependent activation of NF-kB and TGFβ expression. (**A**) Immunofluorescence analysis of L6 cells demonstrates that taurine down-regulates the high levels of phospho-NF-κB expression induced by TNF (Original magnification × 20). (**B,D**) Western Blotting of the NF-κB and IκBα expression. (**C**,**E**) Densitometric analyses were achieved using Stain-free blot to verify equal loading of the samples. (**F**) Real Time PCR analysis of TGFβ expression in L6 cells treated as indicated. Statistical analysis was performed using Student’s *t*-test. * *p* < 0.05; ** *p* < 0.01; *** *p* < 0.01.

**Figure 4 ijms-21-08865-f004:**
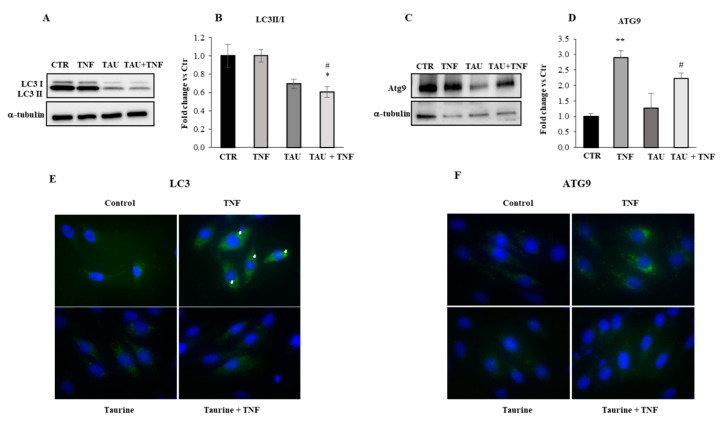
Taurine downregulates the TNF-dependent expression of autophagic markers. (**A**,**C**) Western blot analyses of total lysates were achieved to evaluate the expression levels of LC3 and Atg9 in relation to the treatments indicated in the Figure. (**B**,**D**) The graphs show the densitometric analyses performed by using anti-α-tubulin antibody to verify the loading of the samples. Representative blots are shown. Statistical analysis was performed by Student’s *t*-test. * *p* < 0.05; ** *p* < 0.01; * vs Control, # vs TNF. (**E,F**) Immunofluorescence analysis was performed to evaluate the expression levels and cellular localization of LC3 (head arrows indicate LC3 puncta) and Atg9 (original magnification × 20).

**Figure 5 ijms-21-08865-f005:**
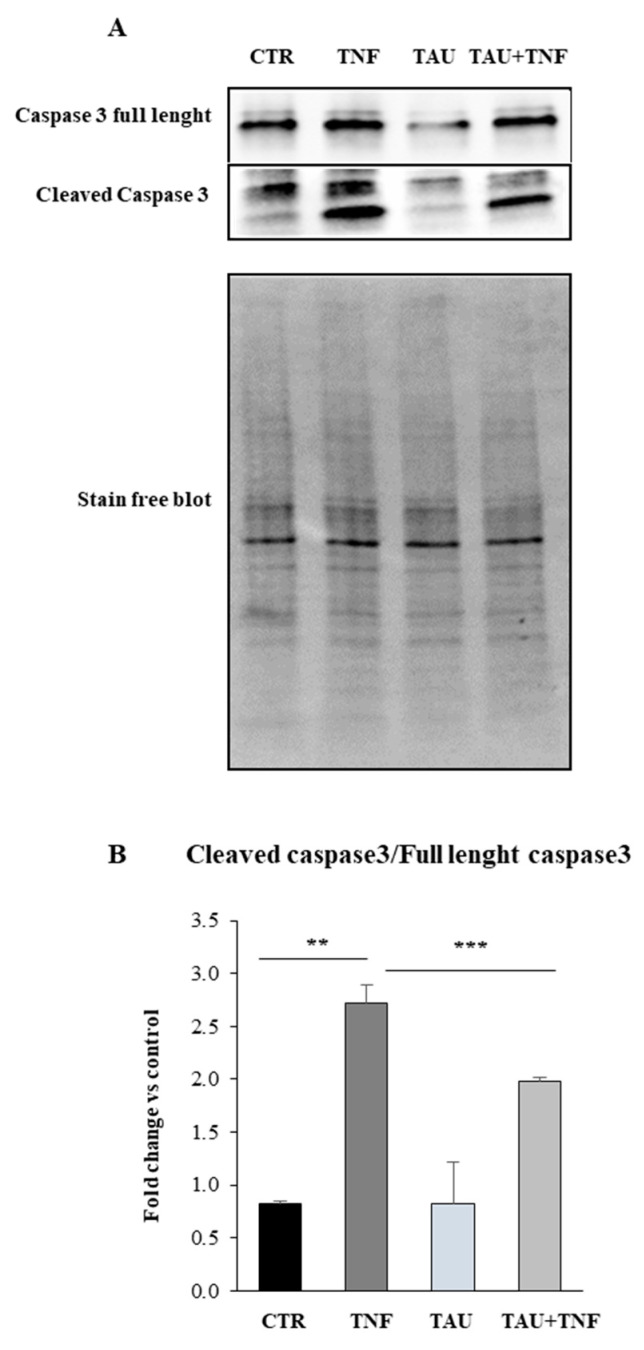
Taurine affects the apoptotic pathway through the modulation of caspase 3 expression. (**A**) Western Blot analysis of caspase full length and the cleaved caspase 3 was achieved in whole extracts of L6 cells after the treatments indicated in the Figure. A representative blot is shown. (**B**) Densitometric analysis performed by using Stain-free blot to verify the loading of the samples. Statistical analysis was performed by Student’s *t*-test. ** *p* < 0.01; *** *p* < 0.01.

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
