# Peer review of "Taurine Attenuates Catabolic Processes Related to the Onset of Sarcopenia"

_ijms, 2020, doi:10.3390/ijms21228865_

Round 1

Reviewer 1 Report

This is an interesting study hypothesizing the protective role of taurine in sarcopenia. This study seeks to evaluate the protective effects of Taurine using an in-vitro system. While this is an interesting study and hypothesis, the enthusiasm was diminished mainly due to some disconnect between presented data and the stated results/conclusions.

Major comments:

  • The study concludes that taurine promotes the myogenic differentiation but the data only presents minimal non-conclusive morphogenic data.
  • The Nfkb western data analysis doesn’t match with the western blotting. The inflammatory profiles need to be complemented with other markers using PCR
  • It is not clear why LC3 autophagy markers are the same in the control and TNF treated cells (Fig 4A). This will make it difficult to conclude if TNF has a different effect on these markers and if TAU treatment has any effects. Also, the western blot band analysis in figure 4D doesn’t match the western blot shown in Fig 4C.
  • It is not clear why alpha-tubulin is the only housekeeping protein used as control and looks missing in Fig 4C. Needs to be checked with GPDH
  • Why the Cleaved Caspase 3 is seen again in both control and TNF treated cells.

Author Response

Major comments:

  • The study concludes that taurine promotes the myogenic differentiation but the data only presents minimal non-conclusive morphogenic data.

The aim of our study was to investigate the effect of taurine in a condition of increased levels of inflammation that typically occurs in sarcopenic muscles. Indeed, the positive effect of taurine on myogenic differentiation has been already reported in the literature (1), so we performed the “morphogenic” experiments to demonstrate that this effect was confirmed in our system being interested in investigating the role of taurine in a condition of altered homeostasis such as increased levels of inflammation induced by TNF treatment. However, in the revised version of our manuscript we included Western Blot analysis of Myosin expression performed after 3 days of treatment indicating that at this time although the myogenic taurine effect is not yet significant compared to the control cultures, taurine is able to counteract the negative effect of TNF on Myosin expression levels. This result is added in Figure 1 and commented in the Results section, page 3, lanes 203-206.

The Nfkb western data analysis doesn’t match with the western blotting. The inflammatory profiles need to be complemented with other markers using PCR.

The NF-kB data analysis reported in Figure 3 refers to different sets of independent experiments (see Original files Fig.3). The Western blot related to NF-kB is a representative image of one of these independent experiments.

As also required by the Reviewer, in the revised version of the manuscript, we included Real-Time PCR analysis of the expression of one of the most relevant inflammatory markers such as TGF-β.  Indeed, it has been reported that TGF-β can synergize with TNF-α or other cytokines suggesting a convergence of the two pathways at common target genes. In particular, NF-κB can be activated by TGF-β and mediate transcription of target genes in a variety of cell types (2). This result is added in Fig.3 and commented in the Results section, pages 5 and 6, lanes 344-360, and in the Discussion section, page 9, lanes 504-507.

  • It is not clear why LC3 autophagy markers are the same in the control and TNF treated cells (Fig 4A). This will make it difficult to conclude if TNF has a different effect on these markers and if TAU treatment has any effects. Also, the western blot band analysis in figure 4D doesn’t match the western blot shown in Fig 4C.

We are grateful to the Reviewer for his observation that allows us to better clarify this point in the revised version of our manuscript. It has to be considered that in our experiments L6 cells were cultured in a serum-free medium that allows us to study the specific role of taurine avoiding potential synergistic or additive effects of factors generally present in the serum. This is particularly relevant for myogenic cells whose differentiation is classically induced by decreasing the concentration of serum in the culture medium (3, 4). Since nutrients are the main regulators of autophagy (5), it is not surprising that this culture condition may induce the autophagy process even in control culture masking the effect of TNF. The point here is the demonstration that taurine can decrease the expression of autophagy markers in this condition.  We rephrased the Results section, page 6, lines 381-384, and introduced these considerations in the Discussion section, page 9, lanes 520-528.

The data analysis reported in Figure 4D refers to different sets of independent experiments (see Original Files Fig.4). Western blotting is representative of only one of these experiments.

  • It is not clear why alpha-tubulin is the only housekeeping protein used as control and looks missing in Fig 4C. Needs to be checked with GPDH.

Alpha-tubulin is one of the markers routinely used in Western Blot analysis of myogenic protein lysates to verify the loading of the sample. In Figure 4C, we probably used a poor quality image that anyway demonstrates how the control sample is more loaded than the others. However, in the data analysis, the expression of the protein of interest is normalized with respect to the quantity of sample loaded. In the revised version of our manuscript, we replaced the image with a higher contrast one hoping that this would satisfy the reviewer's complaint.

  • Why the Cleaved Caspase 3 is seen again in both control and TNF treated cells.

Caspase-3 is a critical executioner of apoptosis ant its activation requires proteolytic processing of its inactive zymogen into activated p17 and p12 fragments. In Figure 5, we demonstrated that the expression of the active form of caspase 3 (cleaved caspase 3) is up-regulated in TNF treated cells, while taurine counteracts the effect of TNF on active caspase 3 expression.

References

  1. Miyazaki, T.; Honda, A.; Ikegami, T.; Matsuzaki, Y. The role of taurine on skeletal muscle cell differentiation. Adv. Exp. Med. Biol. 2013, 776, 321–328.
  2. Luo K. Signaling Cross Talk between TGF-β/Smad and Other Signaling Pathways. Cold Spring Harb Perspect Biol. 2017 Jan 3;9(1):a022137. doi: 10.1101/cshperspect.a022137. PMID: 27836834
  3. Florini JR, Ewton DZ, Magri KA. Hormones, growth factors, and myogenic differentiation. Annu Rev Physiol. 1991;53:201-16. doi: 10.1146/annurev.ph.53.030191.001221. PMID: 2042960.
  4. Florini JR. Hormonal control of muscle growth. Muscle Nerve. 1987 Sep;10(7):577-98. doi: 10.1002/mus.880100702. PMID: 3309650.
  5. Sandri M. Autophagy in skeletal muscle. FEBS Lett. 2010 Apr 2;584(7):1411-6. doi: 10.1016/j.febslet.2010.01.056. Epub 2010 Feb 2. PMID: 20132819.

Reviewer 2 Report

The manuscript by Barbiera et al., describes that Taurine attenuates catabolic processes related to the onset of sarcopenia. This has an important information for the countermeasure of sarcopenia. For the benefit of the reader, however, some points need clarifying. My comments to this article were as follows.

Taurine used in concentration of 200 mM in this manuscript. Why did authors select this concentration?

The immunostainings for DAPI in L6 cells are shown in the middle of Fig. 1D. CTR and TAU+TNF groups will be able to observe the position of DAPI negative cells. Do cells have the influence of combination of TAU and TNF?

In Fig 2A, the amount of protein in the siRNA Akt-transfected taurine treated cells appears low compared with others.

Can we observe that the siRNA Akt-transfected taurine treated cells hardly affect cell differentiation? In general, it has been reported that TNF-alpha downregulates phosphorylation of Akt. How about the levels of mTOR and MEF2 in TNF-alpha treated cells.

The authors show there are the significant differences between CTR and TAU+TNF in Fig. 3C and E. The important findings are comparison of TNF with TAU+TNF.

PI3K/Akt signaling pathway is associated with expression of muscle specific ubiquitin ligases, such as MuRF1 and Atrogin-1. Although we find the result of autophagy as shown in Fig. 4, there are no information of ubiquitin ligase.

In fig. 4C, the amounts of alpha-tubulin without CTR were decreased.

Although we can detect the band of LC3 in CTR in western blotting of Fig. 3A, the fluorescent intensity of LC3 in control group was disappeared. Does the fluorescent intensity of LC3 in TNF group stain in punctate?

The authors examine the effect of taurine on catabolic pathway using undifferentiated muscle cells in Fig. 3, 4, 5. Are differentiated cells (myotubes) the same results as undifferentiated cell?

Lines 305 and 306 were garbled, as 200 M and TNF- .

Lines 322 and 325 were misspell (1,5% and Nf-kB).

Line 445, ….?

Round 2

Reviewer 2 Report

no more comments.